# An Image Encryption Algorithm Based on Complex Network Scrambling and Multi-Directional Diffusion

**DOI:** 10.3390/e24091247

**Published:** 2022-09-05

**Authors:** Yaohui Sheng, Jinqing Li, Xiaoqiang Di, Xusheng Li, Rui Xu

**Affiliations:** 1School of Computer Science and Technology, Changchun University of Science and Technology, Changchun 130022, China; 2Jilin Provincial Key Laboratory of Network and Information Security, Changchun 130022, China; 3Information Center of Changchun University of Science and Technology, Changchun 130022, China

**Keywords:** image encryption, asymmetric encryption, complex network, multi-directional diffusion, chaotic system

## Abstract

Various security threats are encountered when keys are transmitted in public channels. In this paper, we propose an image encryption algorithm based on complex network scrambling and multi-directional diffusion. Combining the idea of public key cryptography, the RSA algorithm is used to encrypt the key related to plaintext. The algorithm consists of three stages: key generation stage, complex network scrambling stage, and multi-directional diffusion stage. Firstly, during the key generation phase, SHA-512 and the original image are used to generate plaintext-related information, which is then converted to plaintext-related key through transformation mapping. Secondly, in the complex network scrambling stage, the chaotic random matrix establishes the node relationships in the complex network, which is then used to construct an image model based on the complex network, and then combines pixel-level and block-level methods to scramble images. Finally, in the multi-directional diffusion stage, the multi-directional diffusion method is used to perform forward diffusion, middle spiral diffusion, and backward diffusion on the image in turn to obtain the final ciphertext image. The experimental results show that our encryption algorithm has a large keyspace, the encrypted image has strong randomness and robustness, and can effectively resist brute force attack, statistical attack, and differential attack.

## 1. Introduction

In this era of rapid development of information technology, a large number of digital images are transmitted on the network. The images contain a wealth of information, for example, the distribution of city buildings and crop growth can be understood in satellite images; medical images can reveal health status, age information, and so on. There are also images that contain sensitive information such as military secrets, commercial information, and government documents. Moreover, when images are transmitted in public networks, security threats such as interception, tampering, and copying may be encountered [1,2,3,4]. Therefore, the image security issue has attracted more and more attention from scholars.

Encryption of images is an effective method of securing images [5,6]. Traditional text encryption schemes are DES [7,8], AES [9], etc., and when using these schemes to encrypt images, it is necessary to first process the image into a bit stream [10], but this ignores the high information content, high redundancy, and high correlation between pixels [11,12,13]. Therefore, traditional schemes have drawbacks when encrypting images and are not suitable for image encryption [14,15]. Therefore, the development of new efficient and secure image encryption algorithms has become an important research topic in the field of security.

Since chaotic systems have characteristics such as initial value and parameter sensitivity, ergodicity, and unpredictability [16,17], these characteristics are well suited for image encryption [18,19]. Many image encryption algorithms based on chaos theory have been proposed by scholars. For example, Hu et al. proposed a coupled chaotic system based on unitary transformation, which can combine two one-dimensional chaotic systems into a chaotic system with better performance, based on which an image encryption algorithm was designed to encrypt the high and low parts of an image separately [20]. Wang et al. propose new spatiotemporal chaotic systems, and design a combination of pixel values, pixel bits, and binary bits in a dynamic hybrid image encryption algorithm [21]. Xiang et al. proposes a method to suppress dynamic degradation of digital chaotic systems and uses the method in image encryption [22]. With the development of technology, scholars have also proposed image encryption algorithms based on the intersection of chaotic systems and compressive sensing [23,24,25], DNA [26,27,28], elliptic curve [29,30], quantum theory [31,32], neural networks [33,34], homomorphic encryption [35,36], and other disciplines. For example, Liu et al. proposed an efficient three-level quantum image encryption scheme based on QArT and Logistic mapping [31]. Shi et al. proposed a visual secure image compression and encryption scheme combining compression awareness and region energy, where the measurement matrix and control pixels are constructed using a hyperchaotic multi-feature system [37]. However, existing chaos-based image encryption algorithms have some drawbacks, some low-dimensional chaos have small key space and narrow chaotic range [38,39], while high-dimensional chaos is highly complex and requires more resources, which increases the computational complexity of the algorithm [40]. Considering the security and computational complexity of the algorithm, in this study, we use a four-dimensional discrete chaotic system to generate chaotic sequences, which was proven to be simple and efficient by Yang et al. [41].

Complex network theory was developed from graph theory in mathematics, which is able to reflect the interactions or relationships within complex systems. Formally, complex networks are composed of nodes and edges. Transportation networks consisting of connected cities and highways and the Internet consisting of computers and networks are examples of complex networks in life. Currently, complex networks have a wide range of applications in social sciences, control engineering, and secure communication [42]. In recent years, more and more researchers at home and abroad have applied complex networks to image processing, using complex network knowledge to construct network models of images and extract the features of images for segmentation, classification, and recognition. Ribas et al. modeled an image as a complex network model and proposed a method for image texture analysis based on the fusion of complex networks and random neural networks [33]. Breve et al. proposed a graph-based interactive image segmentation method that utilizes complex network properties with low time complexity and storage complexity [43]. Wang et al. proposed a method to embed images into complex networks that can fully describe and utilize information at different levels in remote sensing images to mine features of adjacency relationships and apply them to land cover classification [44]. However, existing complex network-based image processing schemes have some drawbacks. For example, when performing network modeling, all pixel points of an image are treated as nodes in a complex network, and this approach increases the complexity of the algorithm and storage space due to the relatively large amount of information contained in the image. In this study, we apply complex networks to the field of data security, mainly for the scrambling part of the proposed algorithm. Simulation results show that complex networks can be effectively used for image scrambling.

Many existing image encryption algorithms use symmetric cryptosystems, which means that the same key is used for encryption and decryption [45,46,47,48]. Therefore, it is difficult to secure the key during encryption and decryption. However, for asymmetric cryptosystems, the public key of the receiver is used for encryption, while the private key corresponding to the receiver is used for decryption. Therefore, asymmetric cryptosystems can solve the security problem of keys in transmission. Scholars have also introduced asymmetric cryptosystems into image encryption algorithms. For example, Liu et al. devised an asymmetric image encryption algorithm that uses the RSA algorithm to encrypt plaintext-related information [49]. Xu et al. designs an asymmetric image encryption scheme suitable for multi-user image transmission [50]. In this study, the key is protected using the Paillier algorithm, which is an asymmetric homomorphic encryption algorithm.

Based on the above discussion, an image encryption algorithm based on complex network scrambling and multi-directional diffusion is proposed in this paper. The algorithm uses a combination of symmetric encryption and asymmetric encryption. Among them, a symmetric encryption method is used for the image, while an asymmetric encryption method is used for the key. In the scrambling phase, pixel-level index scrambling is first performed on the image, followed by block-level scrambling of the image. In block-level disarrangement, a complex network is used to construct an image model, establish the relationship of nodes in the network based on the values of chaotic sequences, and then obtain the upper triangular matrix of the adjacency matrix of the network model, and finally, disarrange the image blocks. In the diffusion stage, forward diffusion, middle spiral diffusion, and backward diffusion are used for the images. The main contributions of this study are as follows:A key protection method based on the RSA algorithm is designed to ensure the security of keys when they are transmitted over a common channel;An image scrambling method based on complex network is designed. When building the network model, the image block is used as the node in the network. This method not only fully scrambles the image, but also reduces the computational complexity and storage space of the algorithm;A multi-directional diffusion method is proposed to spread the diffusion effect from each pixel to the whole image.

The remainder of this study is structured as follows. Section 2 presents the relevant knowledge for this study. In Section 3, the proposed complex network-based scrambling method is described. Section 4 presents the proposed image encryption algorithm. Section 5 evaluates the experimental results and the security of the algorithm. Finally, Section 6 provides a brief summary of the whole paper.

## 2. Relevant Knowledge

### 2.1. A Discrete Four-Dimensional Chaotic System

The chaotic sequences produced by chaotic systems are often used in image encryption schemes using scrambling and diffusion. As the discrete four-dimensional chaotic system used in the literature [41] has good pseudo-randomness and unpredictability, the discrete chaotic system can generate simple and effective chaotic sequences. Therefore, this chaotic system is used in this paper to generate chaotic sequences as key streams. The chaotic system is defined as
(1)$$\left\{\begin{array}{c} x_{1}\left(k+1\right)=sin\left(x_{4}\left(k\right)\right)+sin\left(x_{1}\left(k\right)x_{3}\left(k\right)+x_{1}\left(k\right)+x_{2}\left(k\right)\right)\\ x_{2}\left(k+1\right)=sin\left(x_{2}\left(k\right)\right)-sin\left(x_{1}\left(k\right)\right)\\ x_{3}\left(k+1\right)=sin\left(x_{1}\left(k\right)\right)+sin\left(x_{1}\left(k\right)x_{3}\left(k\right)\right)\\ x_{4}\left(k+1\right)=2sin\left(x_{2}\left(k\right)+x_{4}\left(k\right)\right) \end{array}\right.$$
where $x_{i}\left(i=1,2,3,4\right)$ is the initial state of the chaotic system. All values of variables are limited to [−2,2], we set the initial state of the chaotic system as $(x_{1}\left(0\right),x_{2}\left(0\right),x_{3}\left(0\right),x_{4}\left(0\right))=\left(0.11,0.12,0.13,0.14\right)$. Figure 1 shows the attractor of this discrete four-dimensional chaotic system. Figure 2 is the Lyapunov diagram of the chaotic system; it can be seen that its Lyapunov exponents are 0.3028, 0.0640, −0.5668, −1.7692. Since this chaotic system has two positive Lyapunov exponents, it is more difficult to decipher compared to other chaotic systems. Figure 3 shows the trajectory diagram of this discrete four-dimensional chaotic system. It can be seen from Figure 2 and Figure 3 that this chaotic system has very good chaotic performance and can generate effective pseudo-random sequences. Therefore, this chaotic system is very suitable for image encryption.

### 2.2. RSA Algorithm

The RSA algorithm [51] is an encryption algorithm belonging to a public key cryptosystem. It has a simple encryption process and high security, and is widely used in many security fields. The procedure of the RSA encryption algorithm is described in Algorithm 1.
**Algorithm 1:** RSA algorithm.1:Randomly choose two large prime numbers *p* and *q*, compute n=p∗q and φ(n)=(p−1)∗(q−1).2:Randomly select public key *e*, where 1<e<φ(n) and gcd(e,φ(n))=1.3:Calculate the private key *d*, where e∗d=1modφ(n). Here, {e,n} is the public key, {d,p,q} is the private key.4:The encryption process is $c=m^{e}\left(modn\right)$, and the decryption process is $m=c^{d}\left(modn\right)$.

## 3. Scrambling Method Based on Complex Network

Figure 4 shows a sample diagram of a complex network. The dots in the diagram represent the nodes in the network and the connecting lines in the diagram represent the relationships between the nodes.

In this paper, we adapt the complex network structure of Figure 4 to the network structure shown in Figure 5 in order to use the complex network for image scrambling. The adjustments made are as follows:For an image with M rows and N columns, considering the structure of the image, we restrict the nodes of the complex network to a rectangular space of the same size;Some existing studies use each pixel point of an image as a node of a complex network, but images generally have a large number of pixel count points, and these methods suffer from the problem of large computational effort. In the proposed algorithm, we use image pixel blocks as nodes in the complex network model, which greatly reduces the number of nodes and improves the efficiency of the algorithm;We set the connection probability of a node to ϕ, the degree of that node to α, the total degree of nodes in the network to *s*, and the total number of nodes in the network to κ. Then, the connection probability of each node is
(2)$$\phi =\frac{\alpha +1}{s+\kappa }$$To prevent nodes from being repeatedly connected, the connection probability of a connected node is set to 0;To make random connections of nodes, we use the values of the pseudo-random series generated by the chaotic system and the connection probabilities of the nodes for random connection of nodes.

After generating the complex network model, we obtain the adjacency matrix and the upper triangular matrix of the adjacency matrix of the network nodes from the generated complex network model. In order to reduce the computational effort of the algorithm, here, we traverse the upper triangular matrix of the adjacency matrix to swap the positions of the two nodes with matrix value of 1. Here, we provide an example to explain this part in detail. Figure 6 shows the process of permutation of a matrix MP of size 4×4. The numbers in the figure indicate that the matrix is first chunked. After that, the nodes are randomly connected according to the connection method of the nodes proposed above, then, the adjacency matrix and the upper triangular matrix of this network structure are obtained, and finally, the disordered matrix CMP is obtained by the upper triangular matrix.

## 4. The Proposed Algorithm

In this section, a new image encryption algorithm based on complex network scrambling and multidirectional diffusion is proposed. The encryption algorithm consists of three parts: key generation, image scrambling based on complex networks, and multidirectional diffusion of images, where the last two parts are used to change pixel positions and change pixel values, respectively. The flowchart of the proposed encryption∖decryption scheme is shown in Figure 7. Below, we will describe the encryption process in detail.

### 4.1. Key Generation

As the hashing algorithm is very secure [52], in the proposed algorithm, we use the SHA-512 algorithm to generate the initial key for encryption. The plaintext image is first fed into the SHA-512 algorithm to obtain a sensitive hash value associated with the plaintext image, after which the key used for the dislocation and diffusion parts, respectively, is obtained through this hash value. This method establishes the correlation between the algorithm and the plaintext image, which can resist the chosen plaintext∖ciphertext attack and improve the security of the encryption system. In addition, using different keys for the scrambling and diffusion parts further improves the security of the encryption system. The steps of key generation are as follows:Input the original image *P* of size M∗N into the SHA-512 algorithm to generate a 512-bit hash value *K*.
(3)K=SHA−512(P)The hash value *K* is partitioned into 64 parts, each containing 8 bits of data. Figure 8 shows the partitioning method of the hash value.
(4)$$K=\{K_{1},K_{2},K_{3},\ldots ,K_{63},K_{64}\}$$Use Equation (Equation 5) to calculate the first set of keys $\left\{x_{i}\left(1\right)i=1,2,3,4\right\}$ for the scrambling part of the encryption algorithm.
(5)$$\left\{\begin{array}{c} x_{1}\left(1\right)=\frac{K_{1}⨁K_{2}⨁\cdots ⨁K_{8}}{256}\;\\ x_{2}\left(1\right)=\frac{K_{9}⨁K_{10}⨁\cdots ⨁K_{16}}{256}\;\\ x_{3}\left(1\right)=\frac{K_{17}⨁K_{18}⨁\cdots ⨁K_{24}}{256}\;\\ x_{4}\left(1\right)=\frac{K_{25}⨁K_{26}⨁\cdots ⨁K_{32}}{256} \end{array}\right.$$
where ⨁ represents the XOR operation.Calculate the second set of keys $\left\{x_{i}\left(2\right)i=1,2,3,4\right\}$ using Equation (Equation 6), which is used for the diffusion part of the encryption algorithm.
(6)$$\left\{\begin{array}{c} x_{1}\left(2\right)=\frac{K_{33}⨁K_{34}⨁\cdots ⨁K_{40}}{256}\;\\ x_{2}\left(2\right)=\frac{K_{41}⨁K_{42}⨁\cdots ⨁K_{48}}{256}\;\\ x_{3}\left(2\right)=\frac{K_{49}⨁K_{50}⨁\cdots ⨁K_{56}}{256}\;\\ x_{4}\left(2\right)=\frac{K_{57}⨁K_{58}⨁\cdots ⨁K_{64}}{256} \end{array}\right.$$

For key transmission, we use the RSA algorithm to encrypt the key and ensure that the key is transmitted securely over the public channel. The $\left\{x_{i}\left(1\right)i=1,2,3,4\right\}$ and $\left\{x_{i}\left(2\right)i=1,2,3,4\right\}$ are encrypted into $\left\{c_{i}\left(1\right)i=1,2,3,4\right\}$ and $\left\{c_{i}\left(2\right)i=1,2,3,4\right\}$, respectively, using the Paillier encryption algorithm. The receiver decrypts $\left\{c_{i}\left(1\right)\right\}$ and $\left\{c_{i}\left(2\right)\right\}$ before decrypting the ciphertext image. This method improves the security of the key when it is transmitted over a public network.

### 4.2. Scrambling Process

Based on the existing research, we apply the complex network method to the field of image encryption. Here, the BA scale-free network model is used to construct a model of a planar image and perform image scrambling based on it. The detailed steps in this section are explained below.

Input the first set of keys into the discrete four-dimensional chaotic system, and iterate $M*N+n_{0}$ times to generate four pseudo-random sequences. To eliminate transient effects, the first $n_{0}$ terms of each sequence are discarded, and finally, four pseudo-random sequences $X1_{1}$, $X2_{1}$, $X3_{1}$, $X4_{1}$ of length M∗N are obtained.Sort the sequence $X1_{1}$ from smallest to largest to obtain the index sequence $X1_{1}\_index$.
(7)$$\left[\;,X1_{1}\_index\right]=sort\left(X1_{1}\right)$$Pixel-level scrambling of the original image *P* using the index sequence $X1_{1}\_index$ to obtain the first scrambled image P1. The scrambling method is expressed as
(8)$$P1=P(X1_{1}\_index)$$To further scramble the image and to eliminate the correlation between adjacent pixel blocks, the image is then scrambled by blocks using a complex network. P1 is partitioned into $\left(M/M_{g}\right)*\left(N/N_{g}\right)$ blocks of size $M_{g}*N_{g}$, in this method $M_{g}=N_{g}$. The image blocking method is shown in Algorithm 2.

**Algorithm 2** Image blocking method.**Input:** The image P1 to be blocked, the size of each image block $M_{g}*N_{g}$.
1:[ ,*N*] = size(P1)2:*x* = floor($K_{0}$/*N*) + 13:*y* = mod($k_{0}$,*N*)4:**if** if *y*==0 **then**5:   *x* = *x*− 16:   *y* = *N*7:**end if**8:$P1_{b}\left(K_{0}\right)$ = $P1(M_{g}*\left(x-1\right)+1:M_{g}*x,M_{g}*\left(y-1\right)+1:M_{g}*y)$
**Output:** Image block matrix $P1_{b}$


The initial number of nodes of the complex network model, the number of edges added at each new node, and the final size of the network are set to γ,η, φ. The initial nodes of the network model are set to be isolated. Each node in the network model corresponds to each chunk of the image.In the proposed method, the coordinates of the initial nodes of the complex network are generated using a pseudo-random sequence $X2_{1}$. Firstly, the element values in the sequence $X2_{1}$ are converted to integers between $\left[1,M/M_{g}\right]$, after which two randomly selected sequences of length γ are used as the coordinates of the initial nodes. The generation method is
(9)$$X2_{1xy}=mod\left(floor\left(X2_{1}*10^{15}\right),M/M_{g}\right)$$
where floor() is a downward rounding operation.
(10)$$\left\{\begin{array}{c} n_{x}=X2_{1xy}\left(t_{1}:t_{1}+\gamma -1\right)\\ n_{y}=X2_{1xy}\left(t_{2}:t_{2}+\gamma -1\right) \end{array}\right.$$
(11)$$node=(\left(n_{x}\left(1\right),n_{y}\left(1\right)\right),\left(n_{x}\left(2\right),n_{y}\left(2\right)\right),\ldots ,\left(n_{x}\left(\gamma \right),n_{y}\left(\gamma \right)\right))$$Convert the element values in the sequence $X3_{1}$ to values between (0,1).
(12)$$N\_X3_{1}=mod\left(X3_{1},1\right)$$Generate new nodes, each connected to the previous η nodes, until the total number of nodes in the complex network model reaches φ. Assuming that the degree of a node is *q* and the total degree is $q^{\prime }$, the connection probability of the current node is $\left(q+1\right)/\left(q^{\prime }+1\right)$, which solves the problem that the connection probability of isolated nodes is 0. When making node connections, the nodes are connected randomly using the element values in $N\_X3_{1}$ according to the connection probability of the current node. In this paper, we set γ, η, and φ to 10, 4, and 256, respectively, and Figure 9 shows the graph of this complex network model.Obtain the adjacency matrix *A* and the upper triangular matrix $A^{\prime }$ of the adjacency matrix based on the connection status of the nodes. The final scrambled image P2 is obtained by block scrambling using $A^{\prime }$. The scrambling method is shown in Algorithm 3.

**Algorithm 3** Image block scrambling method.**Input:** Image block matrix $P1_{b}$, upper triangular matrix $A^{\prime }$.
1:[*M*,*N*] = size($A^{\prime }$)2:**for**i=1 to *M* **do**3:   **for** j=1 to *N* **do**4:     **if** $A^{\prime }$ (*i*,*j*) == 1 **then**5:        *a*=$P1_{b}\left\{x\left(i\right),y\left(i\right)\right\}$6:        $P1_{b}\left\{x\left(i\right),y\left(i\right)\right\}=P1_{b}\left\{x\left(j\right),y\left(j\right)\right\}$7:        $P1_{b}\left\{x\left(j\right),y\left(j\right)\right\}=a$8:     **end if**9:   **end for**10:**end for**11:$P2=P1_{b}$
**Output:** Image scrambling result P2

**Figure 9 entropy-24-01247-f009:**
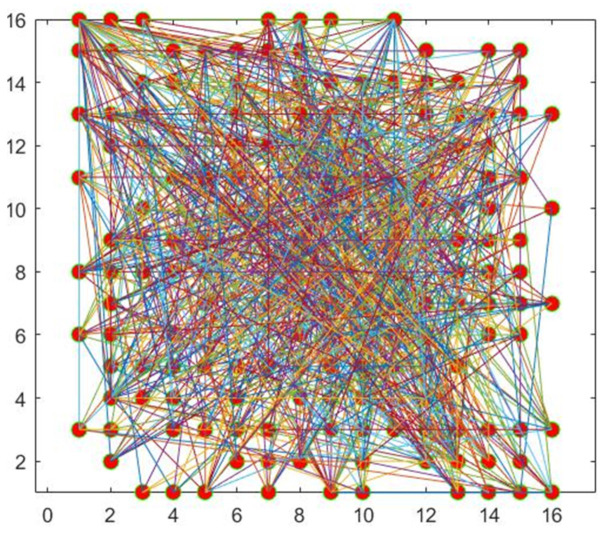
Diagram of the complex network model generated by the described method.

An example is given here to illustrate the proposed scrambling method. We use a “Boat” image of size 256×256, and to explain the proposed method more clearly, we start from step 4. First, divide the “Boat” image into 16 blocks, each of size 64×64. Afterwards, the initial number of nodes of the complex network model, the number of edges added each time a new node is added, and the final scale of the network are set to 4, 4, and 16, respectively. Figure 10 displays the generated adjacency matrix and upper triangular matrix. Figure 11 shows the result of the generated complex network model and image scrambling.

### 4.3. Diffusion Process

An effective encryption algorithm should also have the diffusion property. In the algorithm proposed in this paper, we use a multi-directional diffusion method in order to obtain a good diffusion effect. The method consists of three parts: forward diffusion operating from the top left pixel of the image, spiral diffusion operating from the middle pixel of the image, and backward diffusion operating from the bottom right pixel of the image. To improve the security of the algorithm, a chaotic sequence closely related to the original image is used for each diffusion, while the chaotic sequence used for each diffusion is controlled by the value of the chaotic sequence. The specific diffusion steps are as follows:Input the second set of keys $\left\{x_{i}\left(2\right)i=1,2,3,4\right\}$ into the discrete four-dimensional chaotic system, and also iterate $M*N+n_{0}$ times to generate four pseudo-random sequences. In order to eliminate transient effects, the first $n_{0}$ items of each sequence are discarded as in the previous method, and finally, four pseudo-random sequences $X1_{2}$, $X2_{2}$, $X3_{2}$, $X4_{2}$ of length M∗N are obtained.Convert the values of pseudo-random sequences $X1_{2}$, $X2_{2}$, $X3_{2}$, $X4_{2}$ to integers between [1,255], and the conversion method is as follows
(13)$$\left\{\begin{array}{c} NX1_{2}=mod\left(floor\left(X1_{2}*10^{15}\right),256\right)\\ NX2_{2}=mod\left(floor\left(X2_{2}*10^{15}\right),256\right)\\ NX3_{2}=mod\left(floor\left(X3_{2}*10^{15}\right),256\right)\\ NX4_{2}=mod\left(floor\left(X4_{2}*10^{15}\right),256\right) \end{array}\right.$$Intercept the sequence of length 6 from $NX4_{2}$ and modify the element values of the intercepted sequence to integers between [1,3]. The sequence is divided into three groups, while the elemental values of each group are not the same, and the chaotic sequence used in image diffusion is controlled using the elemental values of one group of the sequence at a time.
(14)$$SNX4_{2}=mod\left(NX4_{2}\left(t:t+5\right),3\right)+1$$Combine the sequence $NX1_{2}$, $NX2_{2}$, $NX3_{2}$ into a matrix.
(15)$$MX=[NX1_{2},NX2_{2},NX3_{2}]$$Perform forward diffusion operation on P2. Starting from the pixel in the top left corner of the image, the diffusion is performed pixel by pixel, from top to bottom, from left to right, ending with the bottom right corner of the image. The next value at the current position is updated with the previous pixel value and two chaotic sequence elements.
(16)$$P3\left(i\right)=\left\{\begin{array}{c} P2\left(i\right)⨁NX4_{2}\left(t+6\right)⨁MX\left(SNX4_{2}\left(1\right)\right)\left(i\right)\\ ⨁MX(SNX4_{2}\left(2\right))\left(i\right)\;for\;i=1\\ P2\left(i\right)⨁P3\left(i-1\right)⨁MX(SNX4_{2}\left(1\right))\left(i\right)\\ ⨁MX(SNX4_{2}\left(2\right))\left(i\right)\;for\;i\neq 1 \end{array}\right.$$Perform a spiral scan operation on P3, starting from the middle pixel of the image and scanning the image in a spiral manner to convert P3 into a one-dimensional matrix $P3^{\prime }$, after which a pixel-by-pixel diffusion is performed starting from the first pixel of $P3^{\prime }$ and ending with the last pixel.
(17)$$P4\left(i\right)=\left\{\begin{array}{c} P3^{\prime }\left(i\right)⨁NX4_{2}\left(t+7\right)⨁MX\left(SNX4_{2}\left(3\right)\right)\left(i\right)\\ ⨁MX(SNX4_{2}\left(4\right))\left(i\right)\;for\;i=1\\ P3^{\prime }\left(i\right)⨁P4\left(i-1\right)⨁MX\left(SNX4_{2}\left(3\right)\right)\left(i\right)\\ ⨁MX(SNX4_{2}\left(4\right))\left(i\right)\;for\;i\neq 1 \end{array}\right.$$Perform a backward diffusion operation on P4. The diffusion is performed pixel by pixel, starting from the pixel in the bottom right corner of the image, going from bottom to top, right to left, and ending with the top left corner of the image. *C* is the final encryption result.
(18)$$C\left(i\right)=\left\{\begin{array}{c} P4\left(i\right)⨁NX4_{2}\left(t+8\right)⨁MX\left(SNX4_{2}\left(5\right)\right)\left(i\right)\\ ⨁MX(SNX4_{2}\left(6\right))\left(i\right)\;for\;i=M*N\\ P4\left(i\right)⨁C\left(i+1\right)⨁MX(SNX4_{2}\left(5\right))\left(i\right)\\ ⨁MX(SNX4_{2}\left(6\right))\left(i\right)\;for\;i\neq 1 \end{array}\right.$$

To better explain this section, we provide an example of a detailed operation on an image of size 4×4. Figure 12 shows the process of the proposed diffusion method, where the numbers in the figure indicate the pixel values of the image.

### 4.4. Decryption Algorithm

After the sender encrypts the original image, the keys $\left\{x_{i}\left(1\right)i=1,2,3,4\right\}$ and $\left\{x_{i}\left(2\right)i=1,2,3,4\right\}$ are encrypted using the public key (n,g) obtained from the receiver and the RSA encryption algorithm. After that, the sender sends the ciphertext image *C* and the encryption keys $c_{i}\left(1\right)i=1,2,3,4$ and $c_{i}\left(2\right)i=1,2,3,4$ to the receiver. The receiver decrypts the keys using the private key (λ,μ) to obtain the initial values of the chaotic system $x_{i}\left(1\right)i=1,2,3,4$ and $x_{i}\left(2\right)i=1,2,3,4$, after which the chaotic sequences $\left\{X1_{1},X2_{1},X3_{1},X4_{1}\right\}$ and $\left\{X1_{2},X2_{2},X3_{2},X4_{2}\right\}$ are obtained by the discrete four dimensional chaotic system. Finally, the decrypted image P is obtained by the inverse process of the diffusion operation and the inverse process of the permutation operation.

## 5. Experimental Results and Performance Analysis

To verify the validity and security of the algorithm, simulations were performed using the Matlab platform on the Window 10 operating system. The selected sample images are grayscale images “Baboon” and “Peppers” with the size of 256×256 and color images “Boat” and “Sailboat” with the size of 256×256×3. Figure 13 shows the results of the simulation. It is clear from the experimental results that the encrypted image is a noise-like image, that it is visually impossible to get any information about the original image from the cipher image, and that a lossless decrypted image can be obtained with the correct key.

### 5.1. Keyspace Analysis

The encryption algorithm must be able to withstand brute force cracking by an attacker [53]. Theoretically, the algorithm is proven to be secure when the key space is larger than $2^{128}$. The key size of the proposed algorithm in this paper is 512 bits and each bit has 2 states. Therefore, the key space size of the proposed encryption algorithm is $2^{512}\left(>>2^{128}\right)$. As a result, the image encryption algorithm proposed can resist the brute force attack.

### 5.2. Key Sensitivity Analysis

Key sensitivity is a reliable way to measure the security of digital image encryption systems [54]. To test the key sensitivity of the proposed algorithm, we randomly modify the key $K_{1}$ generated in the algorithm to obtain two other keys—$K_{2}$ and $K_{3}$. $K_{1}$, $K_{2}$, $K_{3}$ are denoted as

$K_{1}=fe6638a20a69a91f37a8d4374f750206ea27f99e84cc700325631fdac29be7ea67800bdfa$4dcb755cddc9261edd84501418a67a48dce19ef61564407fb6d1435,$K_{2}=fe6638a20a69a91f37a8d4374f750206ea17f99e84cc700325631fdac29be7ea67800bdfa$4dcb755cddc9261edd84501418a67a48dce19ef61564407fb6d1435,$K_{3}=fe6638a20a69a91f37a8d4374f750206ea27f99e84cc700325631fdac29be7ea67800bdfa$4dcb755cddc9261edd84501418a67a48dce19ef61564407fb6d0435.

The results of the key sensitivity analysis of the encryption process of the proposed algorithm are shown in Figure 14. Figure 14b,c show the encryption results using the keys $K_{1}$ and $K_{2}$, respectively. Figure 14d shows the difference of the two encryption results. Figure 15 shows the experimental results of the key sensitivity of the decryption process. It can be seen that although the keys $K_{1}$, $K_{2}$, and $K_{3}$ differ by only one bit, the images produced during decryption are completely different. Figure 15e shows the difference between the decrypted images obtained using illegal keys with only minor differences. The experimental results can show that the proposed image encryption algorithm is highly sensitive to the key.

### 5.3. Information Entropy Analysis

Information entropy can be used to detect the degree of chaos in the distribution of image pixels. The formula for calculating information entropy is [55].
(19)$$H=-\sum_{i=0}^{255}P_{i}log_{2}P_{i}$$
where $P_{i}$ denotes the frequency of occurrence of the pixel with value *i*. The ideal value of information entropy can be obtained when all pixels appear with equal probability. Thus, for an 8-bit image, the theoretical maximum value of information entropy is 8. Table 1 lists Figure 13a–d and the results of information entropy measurements after encryption of these images. From the experimental results, it can be seen that the information entropy of the cryptographic images is very close to the theoretical value, indicating that the proposed algorithm produces cryptographic images with good randomness. Table 2 shows the information entropy results of different algorithms using “Baboon” of size 256×256. In comparison, the encryption result in this paper is closer to the theoretical value. Therefore, the proposed algorithm has better resistance to information entropy analysis.

### 5.4. Histogram Analysis

The histogram tests the distribution of gray values in an image. In general, the original image histogram is fluctuating, while the histogram of an encrypted image is flat [61]. Good encryption algorithms produce cryptographic images with a very uniform distribution of pixel values. Figure 16 shows the results of the histogram analysis. As can be seen from Figure 16, the histograms of all encrypted images are smooth and uniform, indicating that the proposed algorithm hides the original image information and has the ability to resist histogram statistical attacks.

### 5.5. Adjacent Pixel Correlation Analysis

A good image encryption algorithm must be able to completely break the correlation of neighboring pixels in an image [62]. To verify the proposed algorithm, we choose Figure 13a and its cipher image Figure 13e as the test images. Figure 17 shows their adjacent pixel correlation distribution in horizontal, vertical, and diagonal directions. It is clear that the pixels of the original image are mainly distributed around y=x, while the pixels of the cipher image are distributed over the entire coordinate interval.

In addition, it is possible to quantify the correlation between the original and ciphertext images in the three directions of neighboring pixels [63].
(20)$$\left\{\begin{array}{cc} & \;\;\;\;\;\;\;\;\;\;\;\;\;\;\;\;\;\;\;\;\;\;\;\;\;\;r_{xy}=\frac{cov(x,y)}{\sqrt{D(x)D(y)}},\\ cov\left(x,y\right)= & \frac{1}{S}\sum_{i=1}^{S}\left(x_{i}-E\left(x\right)\right)\left(y_{i}-E\left(y\right)\right),\\ & \;\;\;\;D\left(x\right)=\frac{1}{S}\sum_{i=1}^{S}{\left(x_{i}-E\left(x\right)\right)}^{2},\\ & \;\;\;\;\;\;\;\;\;\;\;\;\;\;\;\;\;\;\;\;\;\;\;\;\;\;E\left(x\right)=\frac{1}{S}\sum_{i=1}^{S}x_{i}. \end{array}\right.$$
where *x* and *y* are the grayscale values of two neighboring pixels. Table 3 lists the correlation results for the horizontal, vertical, and diagonal directions of Figure 13a–d. From Figure 17 and Table 3, the pixel distribution of the encrypted image is uniform and the correlation coefficient between adjacent pixels is close to 0. Using the “Boat” image of size 256×256, the adjacent pixel correlation of different encryption algorithms is tested, and Table 4 lists the test results. The proposed method has less correlation compared to other methods and therefore, the encryption algorithm has better resistance to correlation analysis.

### 5.6. Differential Attack Analysis

Differential attacks are a popular type of cryptanalysis technique. The attacker first makes small modifications to the original image, such as changing one or more pixel values, and by analyzing the difference between the two encryption results, the attacker may find a way to break the encryption system. However, the proposed algorithm can effectively resist the differential attack as we use a multi-directional diffusion method, which is able to spread small changes in the image to all pixels. Here, we select “Baboon” as the test image, and the pixel values at the coordinates (100,100) are modified, and Figure 18 shows the test results of the differential attack. Figure 18e shows that the generated cipher image is completely different. The pixel change rate (NPCR) and uniform average change intensity (UACI) can also be used to test the ability of the algorithm to resist differential attacks [64]:(21)$$\left\{\begin{array}{cc} & \;\;\;\;\;\;\;\;\;\;NPCR=\frac{\sum_{i,j}^{}D\left(i,j\right)}{M\times N}\times 100\%,\\ UACI= & \frac{1}{M\times N}\left[\sum_{i,j}^{}\frac{|C_{1}\left(i,j\right)-C_{2}\left(i,j\right)|}{255}\right]\times 100\%, \end{array}\right.$$
where M×N represents the size of the image, $C_{1}$ and $C_{2}$ are the encrypted images obtained by encrypting two images that are only slightly different, and D(i,j) is defined as
(22)$$D\left(i,j\right)=\left\{\begin{array}{c} 0\;C_{1}\left(i,j\right)=C_{2}\left(i,j\right)\\ 1\;C_{1}\left(i,j\right)\neq C_{2}\left(i,j\right) \end{array}\right.$$

Ideally, the values of NPCR and UACI are 99.6094% and 33.4635%, respectively. Table 5 lists the test images using Figure 13a–d as the test images. As can be seen from the table, the test results are very close to the theoretical values. Table 6 compares the experimental results of the proposed encryption algorithm with the existing encryption algorithms for NPCR and UACI, and the selected sample image is a 256×256 “Boat” image. Table 6 shows that the test results of the proposed algorithm are closer to the theoretical values compared to the other algorithms.

### 5.7. Robustness Analysis

Image data may encounter the problem of data loss or interference by noise when stored or transmitted. To verify the robustness of the proposed algorithm against data loss and noise interference, Figure 13e–h are selected as test images, and the images are subjected to clipping and noise interference processing before decrypting these images. Figure 19 and Figure 20 show the test results of clipping processing and noise interference, respectively. The results can show that the proposed algorithm can only affect a small number of pixels when decrypting encrypted images that do not change much, indicating that the algorithm is able to resist noise interference attacks and cropping attacks to some extent.

### 5.8. Encryption Speed Analysis

In this section, the encryption speed of the proposed encryption algorithm is analyzed, and 256 × 256 size “Baboon” images were tested several times to obtain the average value. Table 7 shows the results of the proposed algorithm and other algorithms using images of the same size. As this paper uses a complex network to scramble the image, which takes some time, the proposed algorithm needs a longer time to encrypt the image. However, Table 7 shows that the encryption speed of this algorithm is within an acceptable range.

### 5.9. NIST Test

In this section, we use the NIST test program to test the randomness of chaotic sequences and ciphertext images. This test program includes tests to test the randomness of binary sequences of any length. This paper tests the randomness of chaotic sequences and ciphertext images generated by discrete four-dimensional chaotic systems. The results are shown in Table 8. The results show that the chaotic sequence used in the algorithm and the generated ciphertext image have good randomness.

## 6. Conclusions

In this study, an image encryption algorithm based on complex network scrambling and multi-directional diffusion is proposed. The algorithm uses the asymmetric homomorphic Paillier algorithm to encrypt the sensitive data for generating chaotic sequences. Firstly, two sets of encrypted information associated with plaintexts are generated from the image information, which are used to generate chaotic sequences in the scrambling and diffusion processes, respectively. The scrambling process uses a complex network model as the core, and uses a combination of pixel-level scrambling and block-level scrambling according to the relationship between the nodes in the network model, which fully scrambles the positions of image pixels. The diffusion process uses multi-directional diffusion for forward, middle spiral, and backward diffusion of the image to obtain the cryptographic image. In the process of diffusion, the pixel values are changed with respect to the neighboring pixels and chaotic sequence element values, and the method allows each pixel information to be diffused to the whole image. Simulation experiments show that the method of image dislocation based on the complex network model is feasible. The proposed algorithm is applicable to both grayscale and color images, and the algorithm is resistant to common attacks and has good security. In the future, we will run more tests to verify the efficiency and security of the proposed algorithm, such as testu01 test. Moreover, in addition to the BA scale-free complex network model used in this study, we will study other types of complex network models and design more efficient encryption algorithms to be applied in the field of security for a variety of media files. 

## Figures and Tables

**Figure 1 entropy-24-01247-f001:**
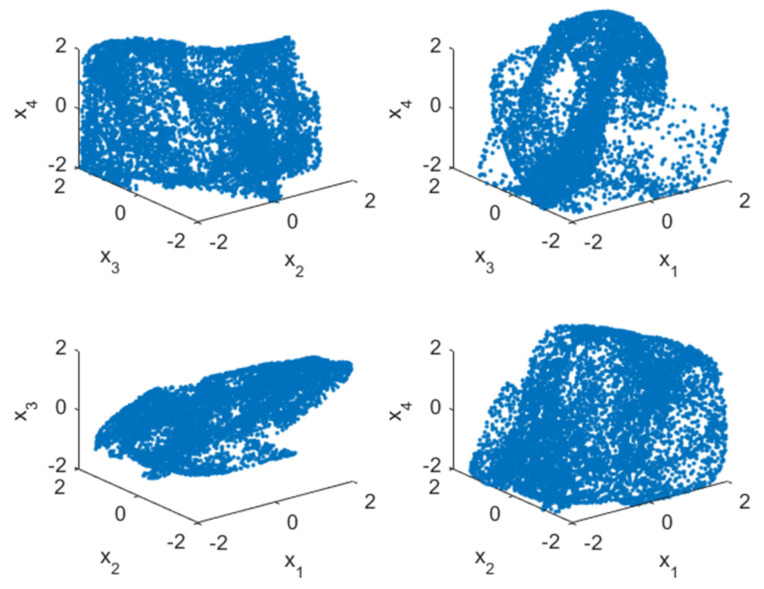
Attractor diagram for discrete four-dimensional hyperchaotic systems.

**Figure 2 entropy-24-01247-f002:**
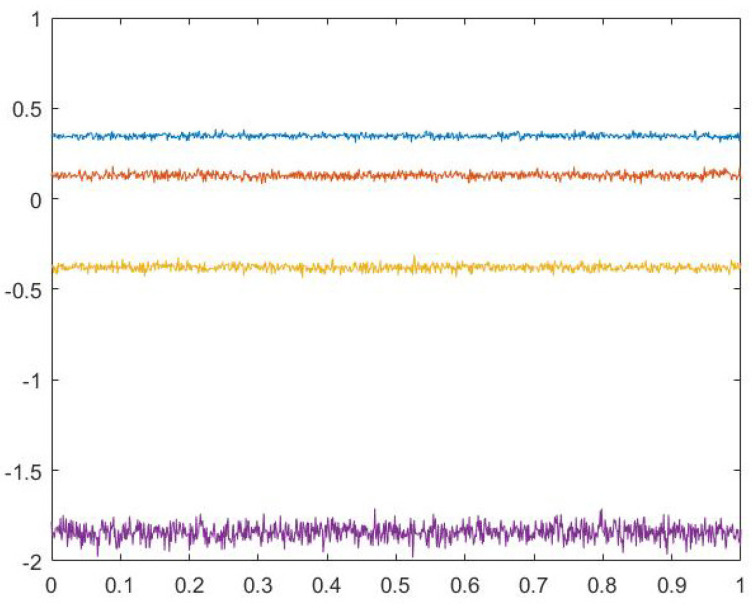
Lyapunov diagrams of chaotic systems.

**Figure 3 entropy-24-01247-f003:**
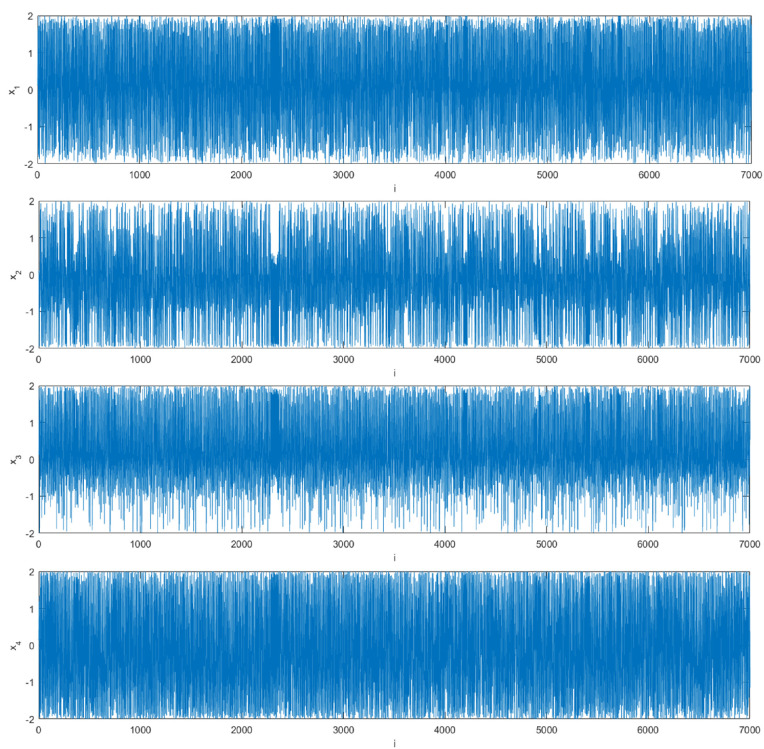
Trajectory diagram of a discrete four-dimensional hyperchaotic system.

**Figure 4 entropy-24-01247-f004:**
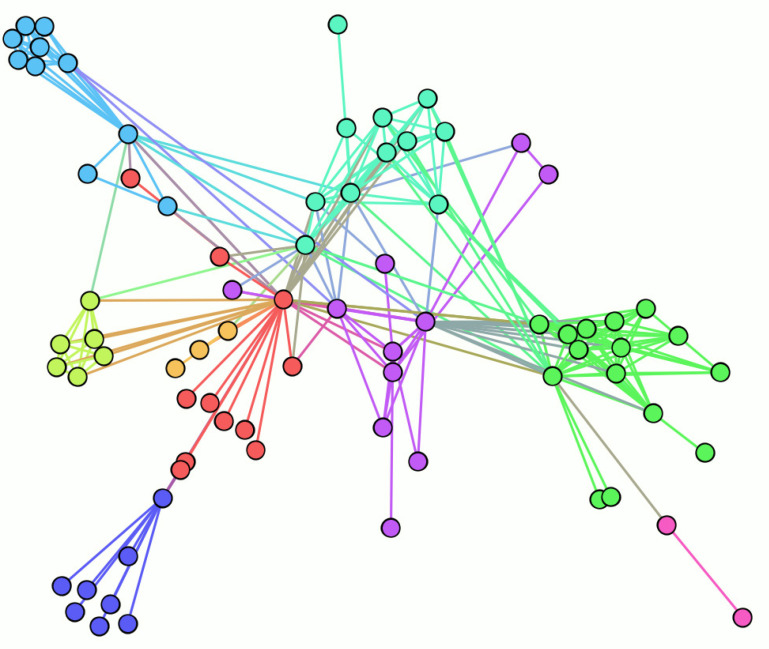
Sample diagram of a complex network.

**Figure 5 entropy-24-01247-f005:**
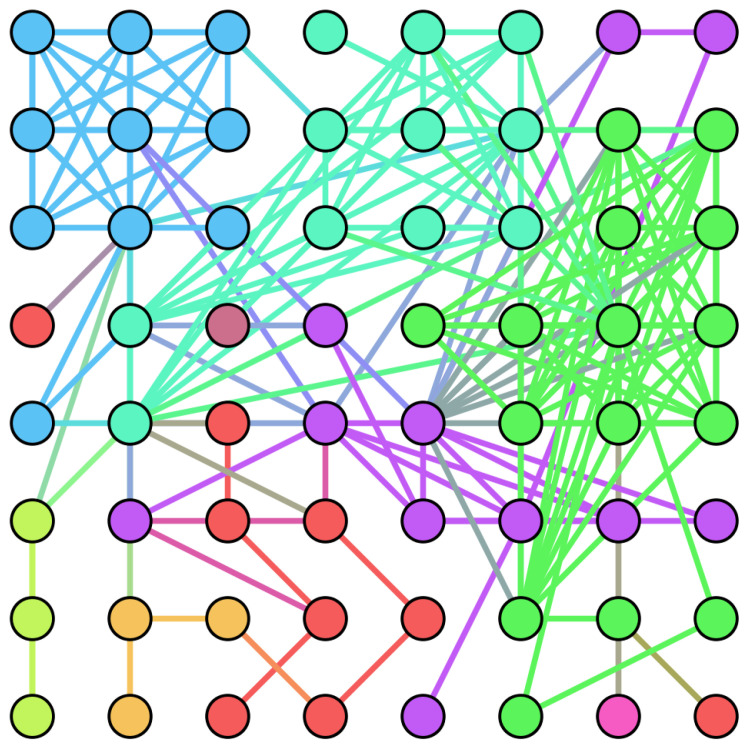
Adjusted complex network diagram.

**Figure 6 entropy-24-01247-f006:**
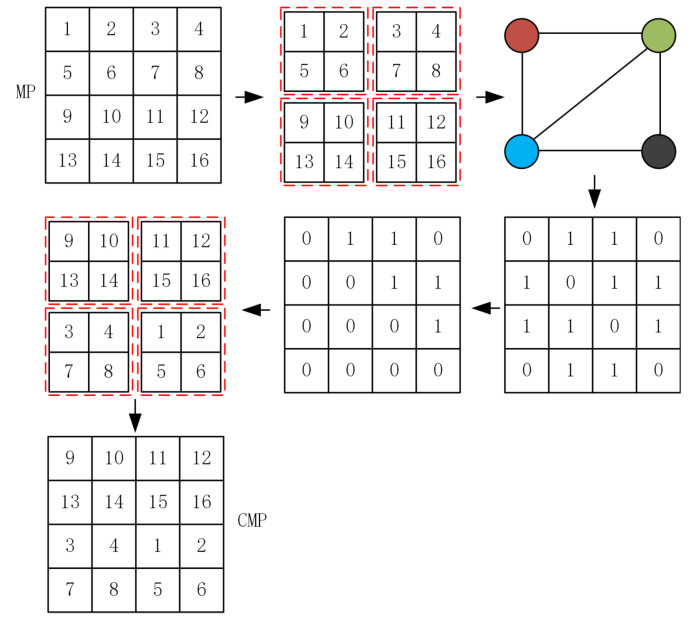
The process of dislocation using complex networks.

**Figure 7 entropy-24-01247-f007:**
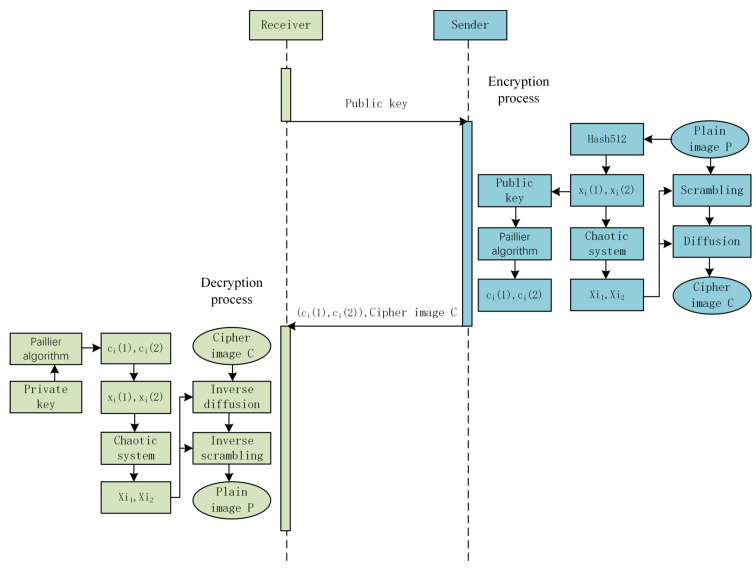
The process of the proposed algorithm.

**Figure 8 entropy-24-01247-f008:**
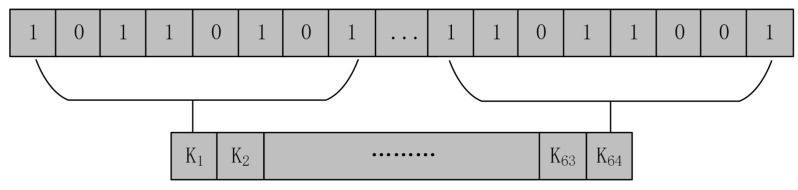
Partitioning method for hash value *K*.

**Figure 10 entropy-24-01247-f010:**
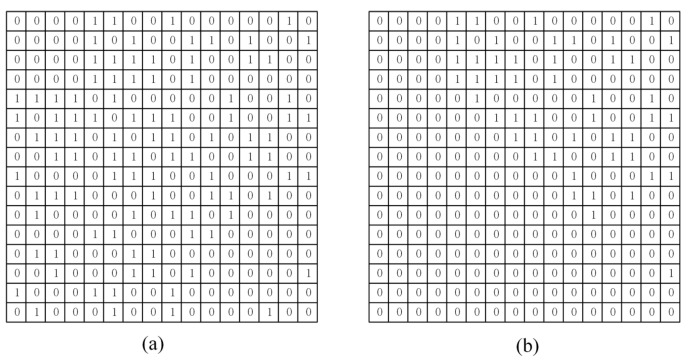
Generated adjacency matrix and upper triangular matrix: (**a**) Adjacency matrix; (**b**) Upper triangular matrix.

**Figure 11 entropy-24-01247-f011:**
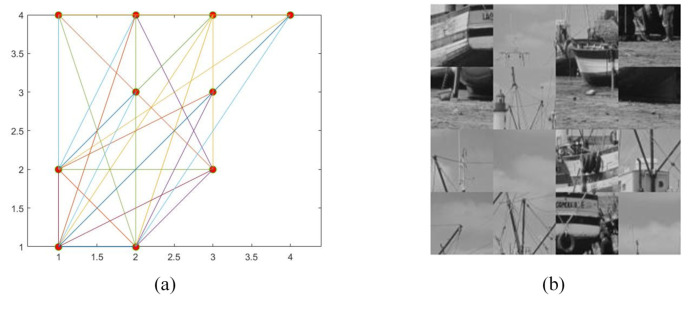
(**a**) The generated complex network model; (**b**) The obtained permutation image.

**Figure 12 entropy-24-01247-f012:**
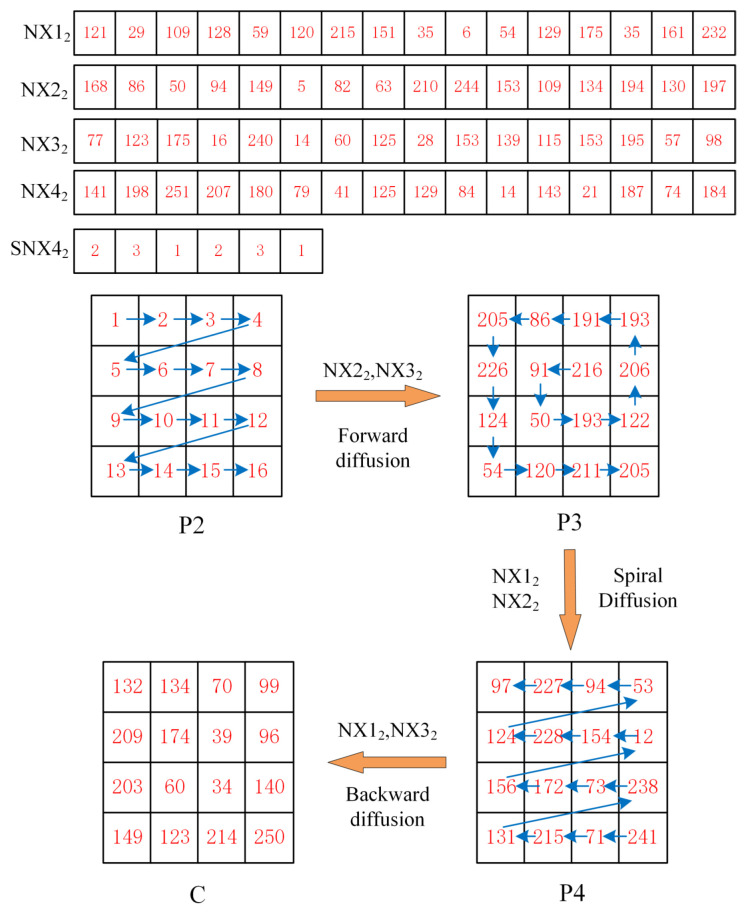
Image diffusion process.

**Figure 13 entropy-24-01247-f013:**
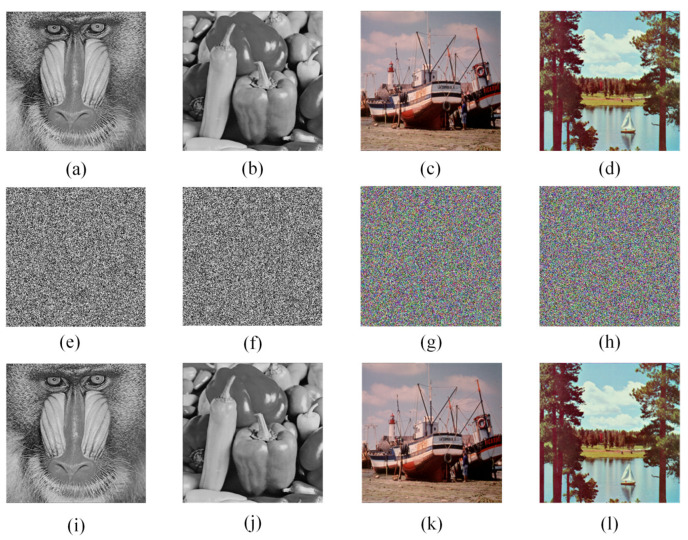
Simulation results: (**a**) Baboon; (**b**) Peppers; (**c**) Boat; (**d**) Sailboat; (**e**–**h**) are the cryptographic images of (**a**–**d**), respectively; (**i**–**l**) are the decrypted images of (**e**–**h**), respectively.

**Figure 14 entropy-24-01247-f014:**
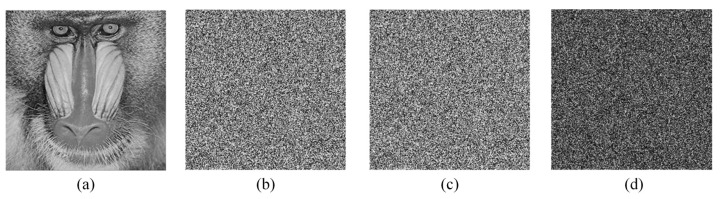
Key sensitivity analysis during encryption: (**a**) Original image *P*; (**b**) Encryption result $C_{1}$ using $K_{1}$; (**c**) Encryption result $C_{2}$ using $K_{2}$; (**d**) Difference between $C_{1}$ and $C_{2}$, |$C_{1}$ − $C_{2}$|.

**Figure 15 entropy-24-01247-f015:**
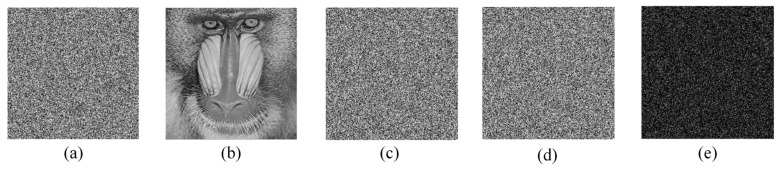
Key sensitivity analysis during decryption: (**a**) Cipher image $C_{1}$; (**b**) Decryption result $D_{1}$ with key $K_{1}$; (**c**) Decryption result $D_{2}$ with key $K_{2}$; (**d**) Decryption result $D_{3}$ with key $K_{3}$; (**e**) Difference between $D_{2}$ and $D_{3}$, |$D_{2}$−$D_{3}$|.

**Figure 16 entropy-24-01247-f016:**
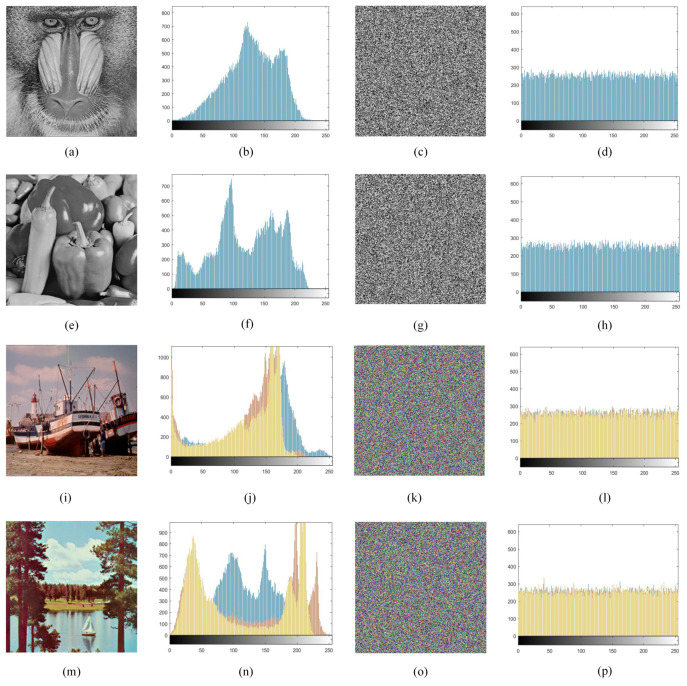
Histogram analysis: (**a**,**e**,**i**,**m**) are the original image; (**b**,**f**,**j**,**n**) are the original image histograms; (**c**,**g**,**k**,**o**) are the cryptographic image; (**d**,**h**,**l**,**p**) are the cryptographic image histograms.

**Figure 17 entropy-24-01247-f017:**
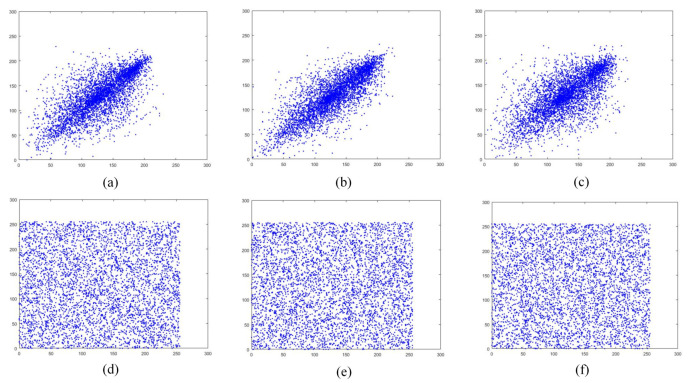
Adjacent pixel correlation: (**a**–**c**) are the adjacent pixel distributions on horizontal, vertical, and diagonal lines of Figure 13e; (**d**–**f**) are the adjacent pixel distributions on horizontal, vertical, and diagonal lines of Figure 13e, respectively.

**Figure 18 entropy-24-01247-f018:**
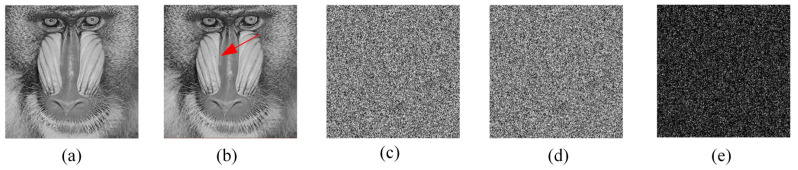
Differential attack analysis: (**a**) Original image $P_{1}$; (**b**) Image $P_{2}$ with one pixel value modified; (**c**,**d**) Encryption results $C_{1}$, $C_{2}$ using the same key pair (**a**,**b**), respectively; (**e**) Difference between $C_{1}$ and $C_{2}$, |$C_{1}$−$C_{2}$|.

**Figure 19 entropy-24-01247-f019:**
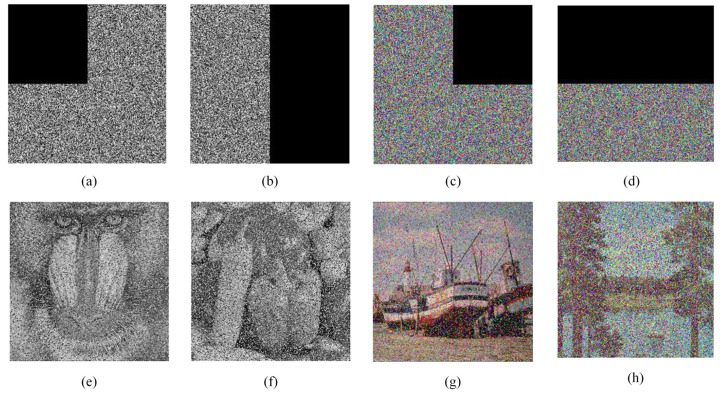
Results of clipping experiments: (**a**,**c**) 25% data loss; (**b**,**d**) 50% data loss; (**e**–**h**) Decrypted images of (**a**–**d**), respectively.

**Figure 20 entropy-24-01247-f020:**
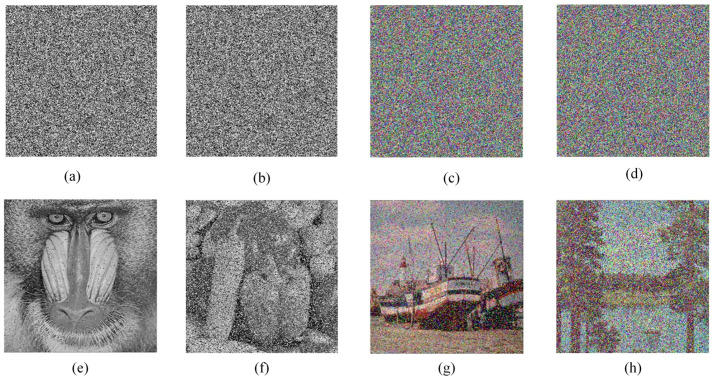
Results of noise interference experiments: (**a**,**c**) with 5% Salt & Pepper Noise; (**b**,**d**) with 10% Salt & Pepper Noise; (**e**–**h**) are the decrypted images of (**a**–**d**), respectively.

**Table 1 entropy-24-01247-t001:** Experimental results of information entropy.

Images	Original Image	Cipher Image
Baboon	7.3500	7.9975
Peppers	7.5739	7.9974
Boat-R	7.4494	7.9972
Boat-G	7.1145	7.9968
Boat-B	7.0501	7.9970
Sailboat-R	7.2673	7.9972
Sailboat-G	7.6262	7.9975
Sailboat-B	7.2006	7.9974

**Table 2 entropy-24-01247-t002:** Information entropy comparison results.

Algorithm	Information Entropy
Our	7.9975
Ref. [56]	7.9973
Ref. [57]	7.9972
Ref. [58]	7.9980
Ref. [59]	7.9974
Ref. [60]	7.9971

**Table 3 entropy-24-01247-t003:** Correlation of adjacent pixels of original image and cipher image.

Image	Original Image			Cipher Image		
	Horizontal	Vertica	Diagonal	Horizontal	Vertical	Diagonal
Baboon	0.9631	0.9292	0.9089	−0.0011	−0.0010	−0.0015
Peppers	0.9680	0.9654	0.9443	0.0005	0.0003	−0.0400
Boat-R	0.9397	0.9283	0.8750	−0.0007	−0.0106	0.0072
Boat-G	0.9481	0.9291	0.8927	−0.0063	−0.0080	0.0028
Boat-B	0.9556	0.9488	0.9115	−0.0073	−0.0019	0.0005
Sailboat-R	0.9518	0.9498	0.9216	0.0006	0.0080	0.0191
Sailboat-G	0.9485	0.9480	0.9138	−0.0078	−0.0011	0.0008
Sailboat-B	0.9612	0.9601	0.9330	−0.0018	0.0006	−0.0174

**Table 4 entropy-24-01247-t004:** Comparison of correlation coefficients of different algorithms.

	Ours	Ref. [56]	Ref. [57]	Ref. [58]	Ref. [59]	Ref. [60]
Horizontal	−0.0011	−0.0352	-	−0.0030	0.0117	-
Vertical	−0.0010	0.0239	-	−0.0090	0.0078	-
Diagonal	−0.0015	−0.0058	-	0.0005	−0.0055	-

**Table 5 entropy-24-01247-t005:** NPCR and UACI test results.

Image	NPCR(%)	UACI(%)
Baboon	99.6063	33.4619
Peppers	99.6047	33.5720
Boat-R	99.6002	33.4270
Boat-G	99.6094	33.4638
Boat-B	99.6078	33.5173
Sailboat-R	99.6215	33.4733
Sailboat-G	99.6033	33.4816
Sailboat-B	99.6048	33.3604

**Table 6 entropy-24-01247-t006:** Comparison results of NPCR and UACI with different algorithms.

	NPCR(%)	UACI(%)
Ours	99.6063	33.4619
Ref. [56]	99.6094	37.9703
Ref. [57]	99.6096	33.4596
Ref. [58]	99.6090	33.4630
Ref. [59]	99.6094	27.9303
Ref. [60]	99.6219	33.5570

**Table 7 entropy-24-01247-t007:** Encryption speed analysis.

	Encryption Time(s)	Encryption Speed (Mbps)
Ours	0.324	1.618
Ref. [6]	3.120	0.017
Ref. [65]	0.703	0.745
Ref. [16]	0.223	2.443
Ref. [62]	0.515	1.018

**Table 8 entropy-24-01247-t008:** NIST test.

Test Name	X1	X2	X3	X4	Encrypted Image
approximate entropy	pass	pass	pass	pass	pass
block-frequency	pass	pass	pass	pass	pass
cumulative sums	pass	pass	pass	pass	pass
FFT	pass	pass	pass	pass	pass
frequency test	pass	pass	pass	pass	pass
linear complexity	pass	pass	pass	pass	pass
long runs of ones	pass	pass	pass	pass	pass
no overlapping	pass	pass	pass	pass	pass
overlapping templates	pass	pass	pass	pass	pass
rank	pass	pass	pass	pass	pass
runs	pass	pass	pass	pass	pass
serial	pass	pass	pass	pass	pass
universal	pass	pass	pass	pass	pass
random excursions	pass	pass	pass	pass	pass
random excursions variant	pass	pass	pass	pass	pass

## Data Availability

Not applicable.

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
