# Peer review of "An Image Encryption Algorithm Based on Complex Network Scrambling and Multi-Directional Diffusion"

_entropy, 2022, doi:10.3390/e24091247_

Round 1

Reviewer 1 Report

1. Tiltle is confusing. If it treated directly the it could be treated as a nonsense.

2. It is not clear why Paillier encryption is used for secret symmetric key agreement? Where homomorphic property is used? It seems that homomorphism is not necessary in this study. Then why this encryption is chosen requiring n^2=p^2*q^2 modulo exponentiation operations while the other asymmetric encryption requires only p^2 modulo exponentiation  operations?

3. Authors are citing DES which is out of date and AES and both are not suitable for their approach.

4. The proposed symmetric encryption scheme depends on the quality of PRNGenerator. The characteristics of proposed generator are not sufficiently investigated. If PRNG is good then encryption will be good as well. Then it is enough to use Vernam cipher. With ideal PRNG the perfect secrecy can be achieved. 

5. Athors statement 29: "Traditional encryption schemes are DES [7,8], AES [9], etc., and when using these schemes to encrypt images, it is necessary to first process the image into a bit stream [10], but this ignores the high information content, high redundancy and high correlation between pixels [11–13]" seems very strange. Encryption including Vernam serves to remove correlation and etc. Moreover, using 64 bits arithmetic bitwise XOR operations are performed very effectively. 

Reviewer 2 Report

In this paper, an image encryption algorithm is proposed. In my understanding, the novelty of this study would be application of the complex network and multi-directional diffusion, which is certainly interesting. The description of the method is clear and the numerical experimental results support the effectiveness of the method. Thus I basically recommend publication.

However, the advantage of this method over other methods has not been sufficiently demonstrated. I suppose that the superiority of the method over a certain existing method should be quantitatively shown. I recommend publication if this issue is addressed.

Reviewer 3 Report

-The interval of the chaotic map (initial conditions and control parameters) should be added to the manuscript.

-Change Figure 1. In order to evaluate the behavior of the map, the plot of Lyapunov exponents of the chaotic dynamical system must be added to the manuscript. 

-The degree of non-periodicity of the proposed scheme should be included in the manuscript. I would suggest the following paper:

 https://doi.org/10.1016/j.cnsns.2013.06.017

-For any cryptosystem, speed and randomness are two primary components. However, none of these have been examined in this manuscript. 

 Speed analysis for evaluating the performance of the algorithm should be included in the manuscript. In the analysis of speed, the relationship between running time and image size should be investigated. Also, the running time should be compared with other published papers.

- The authors should examine the efficiency of the proposed pseudo-random sequences. So the proposed scheme must be subjected to TESTU01 test suites (SmallCrush, Crush, and BigCrush).

- What is the size of plain (input) and ciphered (output) images?

Round 2

Reviewer 1 Report

The corrections were made to avoid misunderstandings.

But nevertheless, it is not clear why Vernam type cipher can not be applied. It should be clarified in the final text.

Author Response

Thank you for raising this point. I'm very sorry that I didn't explain it clearly in my last reply. Strictly speaking, this paper belongs to "Vernam type cipher", because we use the chaotic sequence generated by the chaotic system to encrypt the image, and the length of the generated chaotic sequence is equal to the length of the plaintext image. In addition, we have tested that the generated chaotic sequence has good randomness in the paper, and because the chaotic system has high sensitivity, a little change in the initial conditions can produce completely different chaotic sequences. Therefore, in our algorithm, the chaotic sequence has the same length as the plaintext image, the chaotic sequence has good randomness, and different images can produce completely different chaotic sequences, So we can say that our encryption algorithm belongs to "Vernam type cipher".

Reviewer 3 Report

I have carefully assessed the revised manuscript and your responses to my concerns. I find that the authors are either unable or unwilling to address my major concerns, so I recommend rejection.

Author Response

Thank you for your comments. Your questions are reasonable and we are actively looking for answers. We have added the testu01 test you proposed to the last paragraph of the article as future research content. Thank you for your time and energy in reviewing the manuscript. I wish you good health and smooth work.